# InternLM-XComposer2-4KHD: A Pioneering Large Vision-Language Model Handling Resolutions from 336 Pixels to 4K HD

**Xiaoyi Dong**[*1,2], **Pan Zhang**[*1], **Yuhang Zang**[*1], **Yuhang Cao**[1,2], **Bin Wang**[1], **Linke Ouyang**[1],
Songyang Zhang[1], Haodong Duan[1], Wenwei Zhang[1], Yining Li[1], Hang Yan[1], Yang Gao[1], Zhe Chen[1]
Xinyue Zhang[1], Wei Li[1], Jingwen Li[1], Wenhai Wang[1,2], Kai Chen[1], Conghui He[3], Xingcheng Zhang[3],
Jifeng Dai[4,1], Yu Qiao[1], Dahua Lin[1,2,5], Jiaqi Wang[1,✉]
[1]Shanghai Artificial Intelligence Laboratory, [2]The Chinese University of Hong Kong,
[3]SenseTime Group, [4]Tsinghua University, [5] CPII under InnoHK
`internlm@pjlab.org.cn`

## Abstract

The Large Vision-Language Model (LVLM) field has seen significant advancements, yet its progression has been hindered by challenges in comprehending fine-grained visual content due to limited resolution. Recent efforts have aimed to enhance the high-resolution understanding capabilities of LVLMs, yet they remain capped at approximately $1500 \times 1500$ pixels and constrained to a relatively narrow resolution range. This paper represents InternLM-XComposer2-4KHD, a groundbreaking exploration into elevating LVLM resolution capabilities up to 4K HD ($3840 \times 1600$) and beyond. Concurrently, considering the ultra-high resolution may not be necessary in all scenarios, it supports a wide range of diverse resolutions from 336 pixels to 4K standard, significantly broadening its scope of applicability. Specifically, this research advances the patch division paradigm by introducing a novel extension: dynamic resolution with automatic patch configuration. It maintains the training image aspect ratios while automatically varying patch counts and configuring layouts based on a pre-trained Vision Transformer (ViT) ($336 \times 336$), leading to dynamic training resolution from 336 pixels to 4K standard. Our research demonstrates that scaling training resolution up to 4K HD leads to consistent performance enhancements without hitting the ceiling of potential improvements. InternLM-XComposer2-4KHD shows superb capability that matches or even surpasses GPT-4V and Gemini Pro in 10 of the 16 benchmarks. The InternLM-XComposer2-4KHD model series with 7B parameters are publicly available at `https://github.com/InternLM/InternLM-XComposer`.

## 1 Introduction

In recent years, the progress in Large Language Models (LLMs) (73; 92; 93; 39; 91; 10; 78; 29; 21) has provoked the development of Large Vision-Language Models (LVLMs). These models have demonstrated proficiency in tasks such as image captioning (17; 14) and visual-question-answering (VQA) (57; 31; 33; 107). Nevertheless, due to their limited resolution, they struggle with processing images containing fine details, such as charts (68), tables (87), documents (70), and infographics (69). This limitation constrains their practical applicability in real-world scenarios.

Recent advancements have aimed at enhancing the resolution of Large Vision-Language Models (LVLMs). Some approaches (66; 36; 97; 48) involve adapting high-resolution vision encoders

---

* indicates equal contribution.

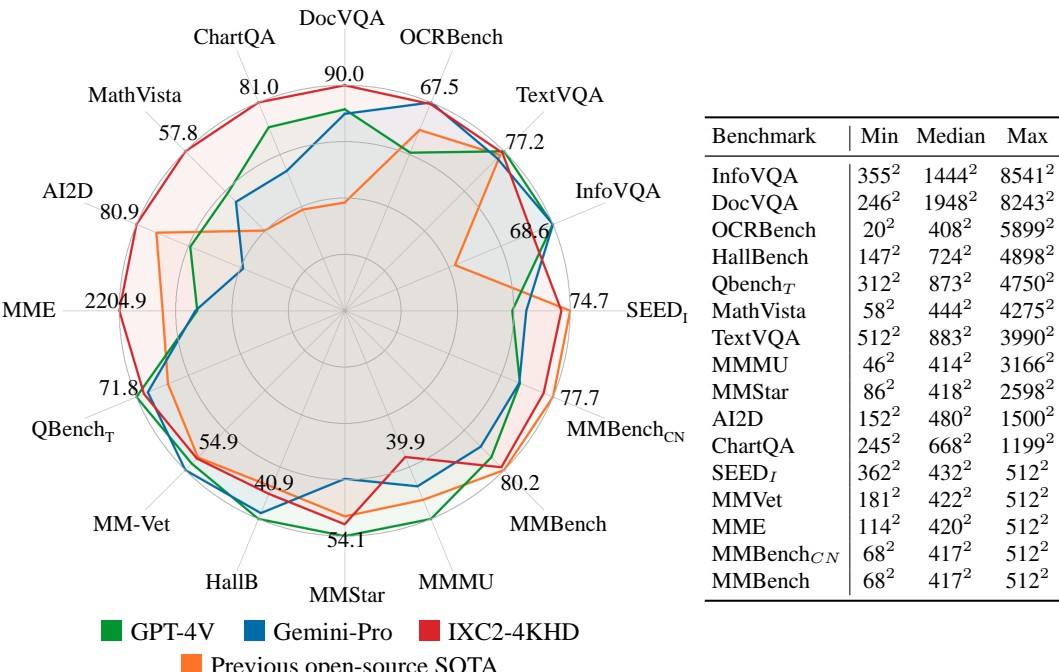

| Benchmark | Min | Median | Max |
|-----------|-----|--------|-----|
| InfoVQA | $355^2$ | $1444^2$ | $8541^2$ |
| DocVQA | $246^2$ | $1948^2$ | $8243^2$ |
| OCRBench | $20^2$ | $408^2$ | $5899^2$ |
| HallBench | $147^2$ | $724^2$ | $4898^2$ |
| $Qbench_T$ | $312^2$ | $873^2$ | $4750^2$ |
| MathVista | $58^2$ | $444^2$ | $4275^2$ |
| TextVQA | $512^2$ | $883^2$ | $3990^2$ |
| MMMU | $46^2$ | $414^2$ | $3166^2$ |
| MMStar | $86^2$ | $418^2$ | $2598^2$ |
| AI2D | $152^2$ | $480^2$ | $1500^2$ |
| ChartQA | $245^2$ | $668^2$ | $1199^2$ |
| $SEED_I$ | $362^2$ | $432^2$ | $512^2$ |
| MMVet | $181^2$ | $422^2$ | $512^2$ |
| MME | $114^2$ | $420^2$ | $512^2$ |
| $MMBench_{CN}$ | $68^2$ | $417^2$ | $512^2$ |
| MMBench | $68^2$ | $417^2$ | $512^2$ |

Figure 1: **(a) Overview of InternLM-XComposer2-4KHD (IXC-4KHD) performance on benchmarks with different resolutions.** Our model based on InternLM2-7B (91) *matches or even surpasses GPT-4V (74) and Gemini Pro (90) in 10 of the 16 benchmarks.* **(b) Image resolution statistic of 16 benchmarks.** We report the minimum (Min), median, and maximum (Max) image area (resolution). Both the inter-/intra-benchmark resolution diversity are large, and we sort them by the maximum resolution.

directly. However, the Vision Transformer (ViT) architecture falls short when dealing with images of varying resolutions and aspect ratios, thereby restricting its ability to handle diverse inputs effectively. Alternatively, some methods (50; 59; 37; 51; 99; 55; 46) maintain the vision encoder's resolution, segmenting high-resolution images into multiple low-resolution patches. Yet, these methods are constrained by an inadequate resolution, typically around $1500 \times 1500$, which does not satisfy the demands of daily content, *e.g.*, website screenshots (85), document pages (70), and blueprints (69). Furthermore, they are confined to either a few predefined high-resolution settings (36; 97; 48; 50; 51; 55; 46; 66; 59) or a limited range of resolutions (101; 37; 99), thereby restricting their utility across a variety of applications.

In this work, we introduce InternLM-XComposer2-4KHD, a pioneering model that for the first time expands the resolution capabilities of Large Vision-Language Models (LVLMs) to 4K HD and even higher, thereby setting a new standard in high-resolution vision-language understanding. Designed to handle a broad range of resolutions, InternLM-XComposer2-4KHD supports images with any aspect ratio from 336 pixels up to 4K HD, facilitating its deployment in real-world contexts.

InternLM-XComposer2-4KHD follows patch division (50; 46) paradigm and enhances it by incorporating an innovative extension: dynamic resolution with automatic patch configuration. To be specific, scaling the resolution of Large Vision-Language Models (LVLMs) to 4K HD and even higher standard is far beyond merely increasing the number of patches. It involves a nuanced approach to overcoming specific challenges: (1) **Dynamic Resolution and Automatic Patch Configuration**: Addressing the scarcity of high-resolution training data, our framework introduces a strategy that dynamically adjusts resolution alongside an automatic layout configuration. During training, it maintains the original aspect ratios of images while adaptively altering patch ($336 \times 336$) layouts and counts. This results in a training resolution that exceeds the original image resolutions, reaching up to 4KHD, addressing the shortfall of high-resolution data. (2) **Handling Variability in Patch Layouts**: Despite the apparent simplicity of dynamic resolution training, the variability in patch layouts can heavily confuse LVLMs. To mitigate this, we introduce a newline token after each row of patch tokens to clearly delineate patch layouts, reducing training ambiguity and significantly boosting performance. (3) **Inference Beyond 4K Resolution:** Our observations reveal that, even when trained on images up to 4K resolution, the model can achieve additional performance improvements during inference by processing images at higher resolutions.

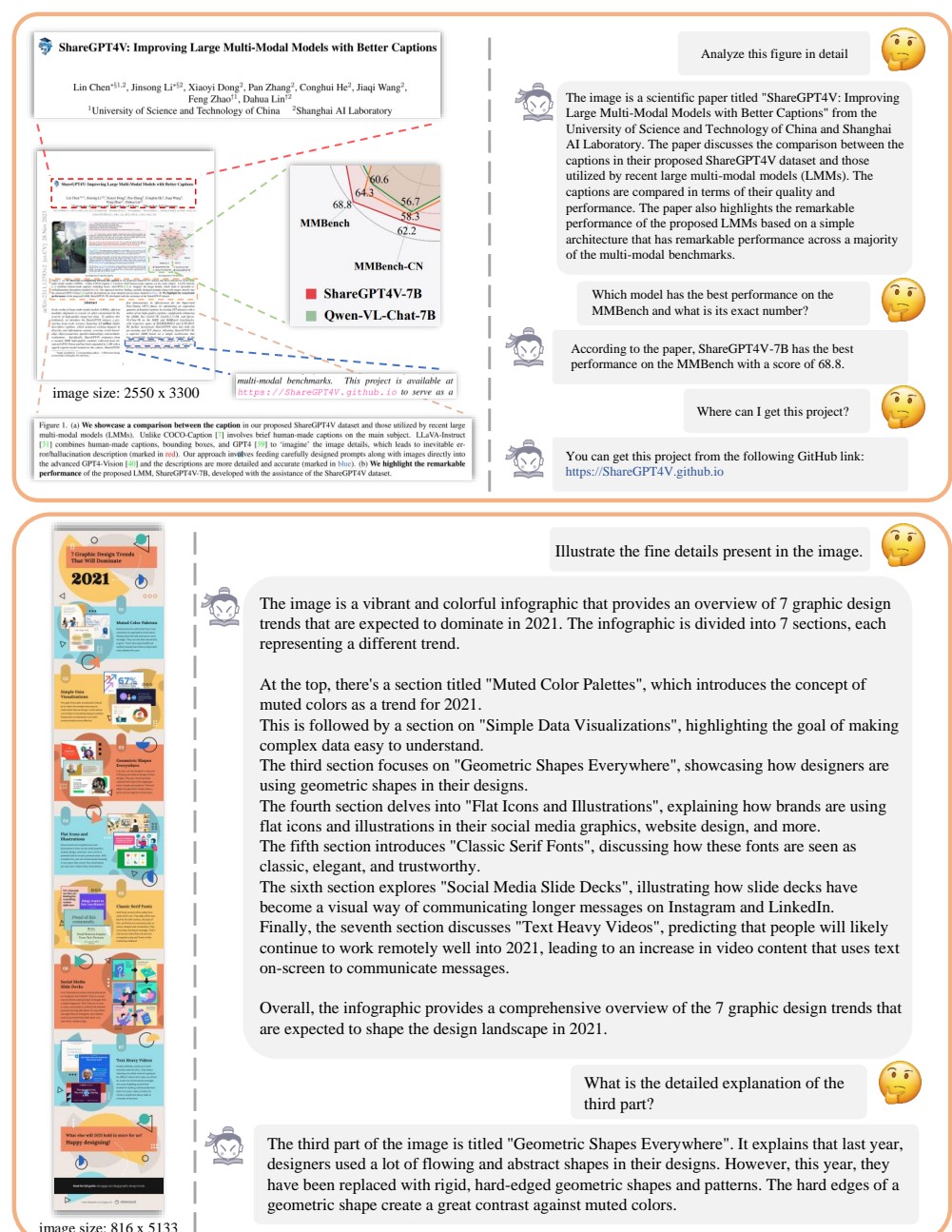

Figure 2: **Chat with InternLM-XComposer2-4KHD**. Some regions of the input HD images are zoomed in for better visualization. For more results please refer to the Supplementary materials.

Furthermore, scaling the training resolution up to 4K standard results in a consistent improvement in performance, highlighting the potential for training even beyond 4K resolution. This underscores the capacity for further enhancing model capabilities and suggests a promising trajectory for advancing the frontiers of high-resolution image processing within the domain of large vision-language models.

We evaluate our InternLM-XComposer2-4KHD on 16 diverse benchmarks spanning various domains, including 5 challenging OCR datasets (InfographicVQA(69), DocVQA(70), OCRBench(58), TextVQA(87), and ChartQA(68)). Compared to previous open-source LVLM models and closed-source APIs, our approach achieves SOTA results in 6 of 16 benchmarks, demonstrating competitive performance despite only 7B parameters. As shown in Figure 1, InternLM-XComposer2-4KHD even surpasses the performance of GPT4V (74) and Gemini Pro (90) across ten benchmarks. Notably, our method exhibits excellent performance on 5 challenging OCR datasets, over existing open-source LVLMs by a substantial margin.

## 2 Related Works

**Large Vision-Language Models (LVLMs).** Large Language Models (LLMs) (9; 76; 73; 23; 41; 92; 93; 39; 91; 108; 6; 78; 10) have gained significant attention due to their impressive performance in various language-related tasks such as text generation and question answering. Following this enthusiasm, recent Large Vision-Language Models (LVLMs) have emerged(74; 19; 16; 18; 28; 32; 113; 25; 110; 7; 47; 77; 102; 4), combining LLMs with vision encoders (79; 109; 89) to leverage the complementary strengths of language and vision modalities. By fusing textual and visual representations, LVLMs can ground language in visual contexts, enabling a more comprehensive understanding and generation of multimodal content (14; 20; 51; 5; 95; 27; 11; 60).

**LVLMs for High-Resolution Understanding.** Large Vision-Language Models (LVLMs) often employ CLIP-ViT as the visual encoder for vision-dependent tasks. However, the visual encoder's reliance on low resolutions, such as $224 \times 224$ or $336 \times 336$ pixels, limits its effectiveness for high-resolution tasks like OCR and document/chart perception. To enhance high-resolution understanding, recent works have primarily employed the following strategies: (1) High-resolution (HR) visual encoders or dual encoders catering to high-resolution (HR) and low-resolution (LR) inputs (66; 97; 36; 48). For instance, Vary (97) introduces a new image encoder supporting HR inputs, which are then concatenated with LR embeddings from the original CLIP visual encoder. Similarly, CogAgent (36) and Mini-Gemini (48) also separate HR and LR images using distinct vision encoders, subsequently merging their features using a cross-attention module. In contrast, our approach offers a more simplified solution and shows advantages for varying resolutions and aspect ratio inputs. (2) Cropped image patches (50; 59; 99; 101; 37; 51; 46). For example, Monkey (50) employs sliding windows to segment images into patches, subsequently processing them with LoRA fine-tuning. TextMonkey (59) further proposes shifted window attention and token resampler to consider the connections among different patches. Fuyu (7) eliminates the need for the image encoder by directly processing a raw image patch sequence. These approaches are confined to either a few predefined high-resolution settings (36; 97; 48; 50; 51; 55; 46; 66; 59) or a limited range of resolutions (37; 99). Conversely, our method devises a dynamic resolution and automatic path configuration strategy to support the scaling from 336 pixels to 4K resolution, and the maximum resolution is larger than previous approaches (*e.g.*, 1.5k for Monkey (50) and 1.2k for UReader (101)). For the first time, our approach discussed the challenges and solutions for handling variability in image feature patch layouts, ensuring effective training with dynamic high resolutions.

**LVLMs for Document Understanding.** Document understanding involves analyzing and comprehending various digital documents, such as figures, tables, and academic papers. Many document understanding tasks require models to handle high-resolution inputs, complex layouts, various aspect ratios, and diverse document formats. To enhance the capabilities of LVLMs for document understanding, several works have collected and constructed high-quality document instruction tuning data, including LLaVAR (112), mPLUG-DocOwl (100) and TGDoc (96). DocPediaDocPedia (30) processes document inputs in the frequency domain. Some previous works have improved document understanding ability by designing special modules for high-resolution inputs, such as HR and LR encoders (36; 97) or cropped image patches (101; 59; 99). Our InternLM-XComposer2-4KHD first scales to 4K resolution inputs and demonstrates strong document understanding ability on OCR-related benchmarks. Also, our approach also achieves comparable results to state-of-the-art open-sourced LVLMs on other general LVLM benchmarks like perception and reasoning (61; 57; 33; 15).

## 3 Method

### 3.1 Model Architecture.

The model architecture of InternLM-XComposer2-4KHD mainly follows the design of InternLM-XComposer2(27) (XComposer2 / IXC2 in the following for simplicity ), including a light-weight Vision Encoder OpenAI ViT-Large/14, Large Language Model InternLM2-7B, and Partial LoRA for efficient alignment.

### 3.2 High-Resolution Input.

**Dynamic Patch Configuration.** Utilizing a static input image size for processing high-resolution images, particularly those with varying aspect ratios, is neither efficient nor effective. To overcome

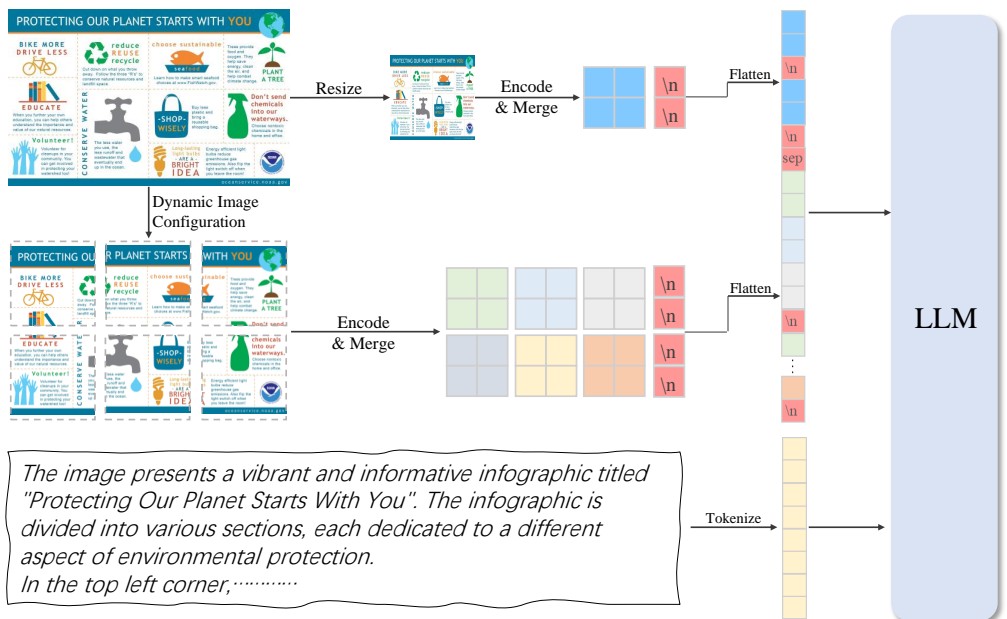

Figure 3: The framework of InternLM-XComposer2-4KHD. Our model processes the high-resolution image with a Dynamic Image Partition strategy and concatenates the image tokens with text tokens as LLM input.

this limitation, we introduce a dynamic patch configuration approach via image partitioning, as shown in Figure 3. Our method strategically segments the image into smaller patches, while maintaining the integrity of the original image's aspect ratio.

Given a maximum patch number $\mathcal{H}$, the image $x$ with size $[h, w]$ is resized and padded to the new image $\hat{x}$ with size $[p_h \times 336, p_w \times 336]$. This process is subject to the following constraints:

$$p_w \times p_h \leq \mathcal{H}; \; p_h = \lceil p_w \times h/w \rceil \tag{1}$$

here $p_w$ and $p_h$ represent the number of patches in each row and column, respectively. We then split the $\hat{x}$ into $p_h \times p_w$ non-overlapped patches. Each patch is a small image with $336 \times 336$ size and we treat these patches as individual inputs for the ViT.

In the following, we use 'HD-$\mathcal{H}$' to represent our high-resolution setting with the constraint of $\mathcal{H}$ patches. For example, the 'HD-9' allows up to 9 patches, including a range of resolutions such as $1008 \times 1008$, $672 \times 1344$, $336 \times 3024$, *etc.*

**Global-Local Format.** For each input image, we present it to the model with two views. The first is the global view, where the image is resized to a fixed size (in our case, $336 \times 336$). This provides a macro understanding of the image. Empirically, we have found this to be crucial for the LVLM to correctly understand the image. The second view is the local view. We divide the image into patches using the previously mentioned Dynamic Image Partition strategy and extract features from each patch. Following feature extraction, the patches are reassembled into a large feature map. The feature map is then flattened to the final local features after a straightforward token merging process.

**Patch Layout Indicator.** Given that an image will have a dynamic patch layout in our method, the number of tokens for each row can vary across different images. This variation will confuse the LVLM, making it difficult to determine which tokens belong to the same row of the image and which ones belong to the next row. This confusion hinders the LVLM's ability to understand the 2D structure of the image, which is crucial for comprehending structural image content such as documents, charts, and tables. To address this issue, we introduce a learnable newline ('\n') token at the end of each row of the image features before the flattening. Finally, we concatenate the global and local views, inserting a special separate ('sep') token between them to distinguish the two views.

### 3.3 Pre-Training

During the pre-training phase, the LLM is frozen while both the vision encoder and Partial LoRA are fine-tuned to align the visual tokens with the LLM following XComposer2(27). The pre-training data

Table 1: **Pre-Training Datasets**. The data are collected from diverse sources for the three objectives.

| Task | Dataset |
|---|---|
| General Semantic Alignment | ShareGPT4V-PT (14), COCO (17), Nocaps (1), TextCaps (86), SBU (75), LAION400M (80), CC 3M (83) |
| World Knowledge Alignment | Concept Data (110) |
| Vision Capability Enhancement | WanJuan (35), Flicker(103), MMC-Inst(54), RCTW-17(84), CTW(106), LSVT(88), ReCTs(111), ArT(22) |

Table 2: **Supervised Fine-Tuning Datasets**. We collect data from diverse sources to empower the model with different capabilities. The image resolution is also different for different tasks.

| Task | Resolution Setting | Dataset |
|---|---|---|
| Caption | HD-25 | ShareGPT4V (14), COCO (17),Nocaps (1) |
| General QA | HD-25 | VQAv2 (3), GQA (38), OK-VQA (67), VD (26), RD(13), VSR(53), |
| Science QA | HD-25 | AI2D (42), SQA (63), TQA(43), IconQA(65) |
| Chart QA | HD-25 | DVQA (40), ChartQA, ChartQA-AUG (68) |
| Math QA | HD-25 | MathQA (104), Geometry3K(62), TabMWP(64), CLEVR-MATH(52)/Super(49) |
| World Knowledge QA | HD-25 | A-OKVQA (81),KVQA (82), ViQuAE(44) |
| OCR QA | HD-25 | TextVQA(87), OCR-VQA(72), ST-VQA(8) |
| HD-OCR QA | HD-55 | InfoVQA(69), DocVQA(70) |
| Conversation | – | LLaVA-150k (56), LVIS-Instruct4V (94), ShareGPT-en&zh (21), InternLM-Chat(91) |

mainly follow the design in XComposer2 which is curated with **three objectives** in mind: 1) general semantic alignment, 2) world knowledge alignment, 3) vision capability enhancement. In this paper, we focus on high-resolution and structural image understanding. So we collected OCR and chart data from diverse sources to enhance this specific capability, as shown in Table.1.

In practice, we employ the OpenAI CLIP ViT-L-14-336 as the vision encoder. We keep the ViT resolution as $336 \times 336$ while adopting Dynamic Patch Configuration to handle higher-resolution images. For pretraining, we use the 'HD-25' configuration, which resizes the input image to a random larger resolution to generate more patches, with the constraint that the total number of patches does not exceed 25. For each image or patch tokens, the image token number is decreased to $1/4$ with a simple **merge operation**. We concatenate the nearby 4 tokens into a new token through the channel dimension, then align it with the LLM by an MLP. The 'sep' and '\n' tokens are randomly initialized. For the Partial LoRA, we set a rank of $256$ for all the linear layers in the LLM decoder block. Our training process involves a batch size of 4096 and spans across 2 epochs. The learning rate linearly increases to $2 \times 10^{-4}$ within the first $1\%$ of the training steps. Following this, it decreases to $0$ according to a cosine decay strategy. To preserve the pre-existing knowledge of the vision encoder, we apply a layer-wise learning rate (LLDR) decay strategy, and the decay factor is set to $0.90$.

### 3.4 4KHD Supervised Fine-tuning (SFT)

After pre-training, we empower the model to understand high-resolution images and solve diverse challenges. Different from conventional perception tasks (*e.g.*, VQAv2, GQA) which typically answer questions based on the noticeable object in the image. OCR-related tasks depend on a detailed understanding of text within a high-resolution image. For instance, in InfoVQA, the length of the longer side of 50% of the images exceeds 2000 pixels. Low-resolution inputs can distort the dense text information, causing the model to fail in its understanding. However, we have observed a resolution saturation phenomena with perception tasks, where a higher resolution makes minor gains.

To address this, we introduce a mixed-resolution strategy for more efficient training. For tasks requiring high resolution, we employ the 'HD-55' setting during training. This allows for the input of 4K ($3840 \times 1600$) images without necessitating additional image compression. These tasks are referred to as the HD-OCR QA tasks in Table 2. For the other tasks, we apply the 'HD-25' resolution setting for them. As in pre-training, we adopt the dynamic-resolution strategy during SFT, images are resized to fall within a range between their original size and the size specified by the 'HD' setting. This dynamic approach enhances the robustness of the LVLM against differences in input resolution, thereby enabling the LVLM to utilize a larger resolution during inference. For instance, we have observed that using the 'HD30' setting yields better results on most OCR-related tasks when the LVLM is trained under the 'HD25' setting.

In practice, we jointly train all the components with a batch size of 2048 over 3500 steps. Data from multiple sources are sampled in a weighted manner, with the weights based on the number of data from each sourced (See Appendix B.1 for more details). As the 'HD-55' setting has double image tokens than the 'HD-25', we adjust the data loader to enable different batch sizes for them and adjust their weight accordingly. The maximum learning rate is set to $5 \times 10^{-5}$, and each component has its

Table 3: **Comparison with closed-source APIs and previous open-source SOTAs.** Our InternLM-XComposer2-4KHD gets SOTA results in 6 of the 16 benchmarks with only 8B parameters, showing competitive results with current closed-source APIs. The best results are **bold** and the second-best results are underlined.

| Method | Doc VQA | Chart QA | Info VQA | Text VQA | OCR Bench | MM Star | Math Vista | AI2D | MMMU | MME | MMB EN | MMB CN | SEED Image | QBench Test | MM-Vet | Hall Bench |
|---|---|---|---|---|---|---|---|---|---|---|---|---|---|---|---|---|
| Open-Source Previous SOTA | (37) 8B 82.2 | (37) 8B 70.2 | (37) 8B 44.5 | (36) 18B 76.1 | (36) 18B 59.0 | (55) 35B 52.1 | (55) 35B 39.0 | (55) 35B 78.9 | (20) 40B 51.6 | (2) 34B 2050.2 | (55) 35B 81.1 | (55) 35B 79.0 | (55) 35B 75.7 | (110) 8B 64.4 | (95) 17B 54.5 | (50) 10B 39.3 |
| *Closed-source API* | | | | | | | | | | | | | | | | |
| GPT-4V | 88.4 | 78.5 | 75.1 | **78.0** | 51.6 | **57.1** | 47.8 | 75.5 | **56.8** | 1,926.5 | 77.0 | 74.4 | 69.1 | **74.1** | 56.8 | **46.5** |
| Gemini-Pro | 88.1 | 74.1 | 75.2 | 74.6 | **68.0** | 42.6 | 45.8 | 70.2 | 47.9 | 1,933.3 | 73.6 | 74.3 | 70.7 | 70.6 | **59.2** | 45.2 |
| IXC2-VL (27) | 57.7 | 72.6 | 34.4 | 70.1 | 53.2 | 55.4 | 57.6 | **81.2** | 41.4 | **2,220.4** | 80.7 | **79.4** | 74.9 | 72.5 | 46.7 | 41.0 |
| IXC2-4KHD | **90.0** | **81.0** | 68.6 | 77.2 | 67.5 | 54.1 | **57.8** | 80.9 | 39.7 | 2,204.9 | 80.2 | 77.7 | 74.7 | 71.8 | 54.9 | 40.9 |

Table 4: **Comparison with open-source SOTA methods.** IXC2-4KHD outperforms competitors in most benchmarks. The best results are **bold** and the second-best results are underlined.

| Method | LLM | MMStar | MathVista | AI2D | $MME^P$ | $MME^C$ | MMB | $MMB^{CN}$ | $SEED^I$ | $QBench^T$ | MM-Vet |
|---|---|---|---|---|---|---|---|---|---|---|---|
| Qwen-VL-Chat | Qwen-7B | 37.5 | 33.8 | 63.0 | 1,487.5 | 360.7 | 60.6 | 56.7 | 58.2 | 61.7 | 47.3 |
| ShareGPT4V | Vicuna-7B | 33.0 | 25.8 | 58.0 | 1,567.4 | 376.4 | 68.8 | 62.2 | 69.7 | - | 37.6 |
| Monkey | Qwen-7B | 38.3 | 34.8 | 62.5 | 1,522.4 | 401.4 | 72.4 | 67.5 | 68.9 | - | 33.0 |
| CogVLM-17B | Vicuna-7B | 36.5 | 34.7 | 63.3 | - | - | 65.8 | 55.9 | 68.8 | - | 54.5 |
| LLaVA-XTuner | InernLM2-20B | - | 24.6 | 65.4 | - | - | 75.1 | 73.7 | 70.2 | - | 37.2 |
| LLaVA-1.5 | Vicuna-13B | 32.8 | 26.1 | 61.1 | 1,531.3 | 295.4 | 67.7 | 63.6 | 68.2 | 61.4 | 35.4 |
| LLaVA-Next | Vicuna-13B | 38.3 | 32.4 | 72.2 | 1,445.0 | 296.0 | 70.0 | 68.5 | 71.4 | - | 44.9 |
| InternLM-XC (27) | InernLM-7B | - | 29.5 | 56.9 | 1,528.4 | 391.1 | 74.4 | 72.4 | 66.1 | 64.4 | 35.2 |
| IXC2-VL | InernLM2-7B | **55.4** | 57.6 | 81.2 | **1,712.0** | 530.7 | 80.7 | 79.4 | 74.9 | 72.5 | 46.7 |
| IXC2-4KHD | InernLM2-7B | 54.1 | 57.8 | 80.9 | 1,655.9 | **548.9** | 80.2 | 77.7 | 74.7 | 71.8 | **54.9** |

own unique learning strategy. For the vision encoder, we set the LLDR to 0.9, which aligns with the pretraining strategy. For the LLM, we employ a fixed learning rate scale factor of 0.2. This slows down the update of the LLM, achieving a balance between preserving its original capabilities and aligning it with vision knowledge. It takes almost 40 hours with 256 A100 GPUs.

## 4 Experiments

In this section, we validate the benchmark performance of our InternLM-XComposer2-4KHD (IXC2-4KHD in the following for simplicity) after supervised fine-tuning.

### 4.1 LVLM Benchmark results.

In Table 3 and Table 4, we compare our IXC2-4KHD on a list of benchmarks with both SOTA open-source LVLMs and closed-source APIs. Here we report results in DocVQA(70), ChartQA(68), InfographicVQA(69), TextVQA(87), OCRBench(58), MMStar(15), MathVista(61), MMMU(107), AI2D(42), MME (31), MMBench (MMB) (57), MMBench-Chinese (MMB$^{CN}$) (57), SEED-Bench Image Part (SEED$^I$)(45), QBench-Testset (QBench$^T$)(98), MM-Vet (105), HallusionBench (HallB)(34). The evaluation is mainly conducted on the OpenCompass VLMEvalKit(24) for the unified reproduction of the results.

**Comparison with Closed-Source APIs.** As demonstrated in Table 3, IXC2-4KHD exhibits competitive performance across a variety of benchmarks, rivaling that of Closed-Source APIs. Owing to its high-resolution input, IXC2-4KHD achieves a score of 90.0% on DocVQA and 81.0% on ChartQA, thereby surpassing GPT-4V and Gemini-Pro with a non-trivial margin. In the challenging InfographicVQA task, our model is the first open-source model that is close to the performance of Closed-Source APIs, exceeding the performance of previous open-source models by nearly 20%. In addition to OCR-related tasks, IXC2-4KHD is a general-purpose Large Vision-Language Modal that excels in semantic-level tasks, demonstrating competitive results.

**Comparison with Open-Source Models.** We also conduct a comprehensive comparison with open-source LVLMs under a similar model scale. As shown in Table 4, our model significantly outperforms existing open-source models, achieving competitive results across all benchmarks.

**High-resolution Understanding Evaluation.** Then we compare IXC2-4KHD with models that are specifically designed for high-resolution understanding tasks. We report the results of 5 high-resolution benchmarks in Table 5, as a general LVLM, IXC2-4KHD shows superb performance on

Table 5: **High-resolution Evaluation.** IntenrLM-XComposer2-4KHD has the largest input resolution and outperforms open-source LVLMs which are specifically tuned for document understanding.

| Model | Model Size | Max Resolution | DocVQA$^{Test}$ | ChartQA$^{Test}$ | InfoVQA$^{Test}$ | TextVQA$^{Val}$ | OCRBench |
|---|---|---|---|---|---|---|---|
| TextMonkey(59) | 9B | 896x896 | 73.0 | 66.9 | 28.6 | 65.6 | 55.8 |
| LLaVA-UHD (99) | 13B | 1008x672 | — | — | — | 67.7 | — |
| CogAgent (36) | 17B | 1024x1024 | 81.6 | 68.4 | 44.5 | 76.1 | 59.0 |
| UReader (101) | 7B | 1120x896 | 65.4 | 59.3 | 42.2 | 57.6 | — |
| DocOwl 1.5 (37) | 8B | 1344x1344 | 82.2 | 70.2 | 50.7 | 68.6 | — |
| IXC2-4KHD | 8B | 3840x1600 | **90.0** (+7.8) | **81.0** (+10.8) | **68.6** (+17.9) | **77.2** (+1.2) | **67.5** (+8.5) |

Table 6: **Influence of Inference Resolution.** The model achieves better performance on text-related tasks when the inference resolution is higher than its training resolution.

| Train | Eval | Doc | Info | Text | Chart | MMB | MME | SEED* |
|---|---|---|---|---|---|---|---|---|
| HD9 | HD9 | 79.4 | 50.5 | 73.8 | 78.2 | 79.5 | 2,201 | 76.6 |
| | HD16 | 83.0 | 58.6 | 74.3 | 75.8 | 79.3 | 2,198 | 76.7 |
| HD16 | HD16 | 84.9 | 60.8 | 75.7 | 80.1 | 80.2 | 2,129 | 75.7 |
| | HD25 | 85.9 | 62.1 | 75.8 | 79.1 | 80.1 | 2,100 | 75.4 |
| HD25 | HD25 | 87.0 | 63.6 | 76.0 | 80.3 | 78.5 | 2,209 | 74.9 |
| | HD30 | 87.4 | 64.6 | 76.2 | 79.4 | 78.9 | 2,173 | 74.3 |

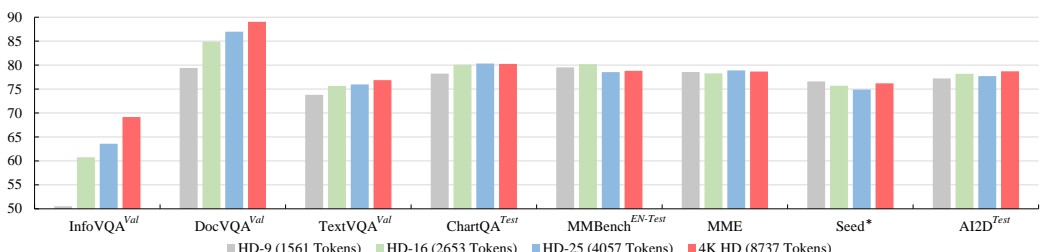

Figure 4: **Influence of Training Resolution.** High-resolution training is critical for HD-OCR tasks, while its gain on other tasks is minor.

these tasks and outperforms competitors with a large margin. For example, IXC2-4KHD gets 68.6% on InfographicVQA, surpassing recent DocOwl 1.5 with +17.9%. For the OCRBench, IXC2-4KHD gets 67.5%, outperforms CogAgent with +8.5%.

## 4.2 Dive into Resolution

**High-Resolution Training is Critical for HD-OCR tasks.** We study four resolution settings: HD-9 (1561 image tokens at most, we simply the statement in the following), HD-16 (2653 tokens), HD-25 (4057 tokens), and 4KHD (8737 tokens). Here we report the validation set of InfoVQA, DocVQA, and TextVQA, test set of ChartQA and AI2D, MMBench EN-Test, and a 2k subset of SEEDBench (we denote it as SEED*). In the following, we report results on the above benchmarks by default.

As illustrated in Fig.4, we note a significant improvement in the HD-OCR tasks as the resolution increases. For instance, the model achieves only a 50.5% score on the InfographicVQA with the HD-9 setting. However, when we switch to the HD-16 setting, we observe a performance gain of +10.2%. The performance continues to improve as the resolution increases, with saturation not observed even for the 4KHD setting. Due to computational constraints, we defer the exploration of the upper bound of improvement to future work. In terms of other OCR-related tasks, the performance gain attributable to increased resolution is relatively minor. For the perception-related benchmarks, performance is saturated on the resolution that only has negligible difference between the four settings.

**Higher Inference Resolution Leads to better results on Text-related Tasks.** An intriguing observation from our experiments is that our model, when inferring with a higher resolution, tends to yield improved results on text-related tasks. We present the results of HD-9, HD-16, and HD-25 in Table 6. For instance, IXC2-HD9 achieves a 50.5% score on InfographicVQA. When we infer with HD16, we see a performance gain of +8.1%, without additional training. Similar improvements are also observed with IXC2-HD16 and IXC2-HD25. We posit that the dynamic image token length used in training enhances the robustness of the LVLM, leading to better results when the text in the image is more 'clear' in the higher-resolution input. Conversely, the results on ChartQA consistently degrade under this setting. This could be due to the model becoming confused about the chart structure when

Table 7: **(a) Influence of Indicator '\n' in the Image Features.** '\n' helps LVLM understand structural images when the input resolution is dynamic and large. **(b) Ablation on Token Merging Operation.** Both the simple concatenation operation and the C-Abstractor works well.

| Model | '\n' | Doc | Info | Text | Chart | MMB | MME | SEED* |
|---|---|---|---|---|---|---|---|---|
| HD9 | × | 79.5 | 50.3 | 74.0 | 78.2 | 79.1 | 2206 | 75.9 |
| HD9 | ✓ | 79.4 | 50.5 | 73.8 | 78.2 | 79.5 | 2201 | **76.6** |
| 4KHD | × | 88.1 | 67.4 | 75.9 | 80.4 | 79.9 | **2232** | 76.4 |
| 4KHD | ✓ | **89.0** | **69.3** | **77.2** | **81.0** | **80.2** | 2205 | 76.2 |

| Strategy | Doc | Info | Text | Chart | MMB | MME | SEED* |
|---|---|---|---|---|---|---|---|
| Re-Sampler | 86.2 | 67.1 | 75.3 | 78.8 | 79.6 | 2124 | 74.2 |
| C-Abstractor | 88.6 | 69.5 | 77.1 | 80.6 | 80.4 | 2236 | 76.7 |
| Concat | 89.0 | 69.3 | 77.2 | 81.0 | 80.2 | 2205 | 76.2 |

Table 8: **Influence of Global-View in the Input.** Global-view is critical for most benchmarks.

| Model | Doc | Info | Text | Chart | MMB | MME | SEED* |
|---|---|---|---|---|---|---|---|
| HD9 | 79.4 | 50.5 | 73.8 | 78.2 | 79.5 | 2201 | 76.6 |
| + w/o global-view | 78.1 | 47.9 | 71.2 | 77.9 | 75.1 | 2019 | 76.2 |

Table 9: **Strategy-Level comparison between LLaVA-Next and our IXC2-4KHD.** Our strategy reaches better performance under a similar image token number constrain.

| Model Nmae | HD Strategy | Image Tokens | Max Resolution | DocVQA | TextVQA | ChartQA | MMBench |
|---|---|---|---|---|---|---|---|
| LLaVA-Next | LLaVA-Next | 2880 | 672x672 | 78.2 | 52.0 | 69.5 | 72.1 |
| IXC-LLaVA | LLaVA-Next | 2880 | 672x672 | 78.9 | 71.5 | 74.1 | 77.6 |
| IXC-4KHD | HD9 | 1440 | 1008x1008 | 79.4 | 73.8 | 78.2 | 79.5 |
| IXC-4KHD | HD16 | 2448 | 1344x1344 | 84.9 | 75.7 | 80.1 | 80.2 |

the resolution is increased. Additionally, similar to the observation from Figure 4, the impact of resolution on perception-related benchmarks appears to be quite minor.

## 4.3 High-Resolution Strategy Ablation

**The Role of Global-View.** We first examine the impact of the global view in our Global-Local Format. As indicated in Table 8(a), we find that the global view is essential for the LVLM to accurately comprehend the input image. When it is removed, the model performs worse across all benchmarks. For instance, the model experiences a $-4.4\%$ drop in performance on the MMBench EN-Test without the global view. We contend that the global view offers a general macro understanding of the image, which the model struggled to derive from the large number of tokens in the local view.

**The Role of the Newline Token.** We incorporate a special newline token at the end of each row of the image features before the flattening operation. This token serves as an indicator of the image's 2D structure. We examine its impact on both the HD-9 and 4KHD strategies in Table 7(a). When a fixed high-resolution strategy HD-9 is employed, we observe that the benefit derived from the newline token is minor. This could be attributed to the LVLM's ability to handle limited differences in image ratios after training. However, when we implement a more challenging 4KHD (HD-25 + HD-55) strategy, which exhibits significant diversity in both image ratio and token number, the LVLM demonstrates a notable decline in performance on OCR-related tasks without the newline indicator. This finding supports our hypothesis that the LVLM struggles to comprehend the shape of the image when the image tokens are directly flattened into a 1D sequence. The newline token can assist the model in better understanding the structure of the image.

**Influence of Token Merging Operation.** In practice, we employ a simple merging operation that concatenates four adjacent tokens along the channel dimension. We have found this approach to be effective in reducing the number of image tokens efficiently. Here we study the influence of different token-merging operations under the 4KHD setting. In Table 7(b), we study two additional strategies: Re-Sampler(5) and C-Abstractor(12), with their default setting and the same compressing rate $0.25$, *i.e.*, reducing an image with 576 tokens to 144 tokens. Results show that both concatenation and C-Abstractor work well and get similar results on most benchmarks, this observation is also consistent with the study in MM-1(71) that the influence of the connector is minor. However, the Re-Sampler performs worse than the other methods with a noticeable margin. We argue this is caused by the learnable queries used for gathering information requiring a great number of data for training, our pre-training data is somewhat lightweight for it to converge fully.

Table 10: **Inference Efficieny Analysis.** The image token number mainly inference the prefix speed, and their difference in the decoding part is neglectable.

| HD | image tokens | prefix encoding time | per-token decoding speed | time to generation 2048 new tokens |
|------|--------------|----------------------|--------------------------|-------------------------------------|
| HD9  | 1440         | 0.2845               | 0.0982                   | 201.4                               |
| HD16 | 2448         | 0.3966               | 0.0983                   | 201.7                               |
| HD25 | 3744         | 0.5513               | 0.0981                   | 201.5                               |

**Strategy-Level Comparison.** For a fair comparison, we trained a new model with the identical architecture, training strategy, and dataset as our original model, but with one key modification: we adopted the high-resolution strategy from LLaVA-Next. We name it as IXC-LLaVA and compare it with 1) LLaVA-Next 0530 official results and 2) IXC2-4KHD under HD-9/16 setting. The results in Table.9 demonstrate that IXC-LLaVA achieves promising performance across all six benchmarks, leveraging the benefits of additional training data and advanced IXC architecture design. However, it is outperformed by IXC2-4KHD HD-9, which utilizes fewer image tokens yet yields better results. This fair comparison underscores the efficiency and effectiveness of our proposed high-resolution strategy, highlighting its advantages over the LLaVA-Next approach.

**Inference Efficiency Analysis.** Our model processes high-resolution images with numerous image tokens, and here we study its inference efficiency in real-world usages. The model inference process consists of two stages: encoding the prefix (model input) and autoregressively decoding new tokens (model output). Correspondingly, the inference efficiency considers two parts: time to encode the prefix and speed to decode each token. Here we report the prefix encoding time and per-token decoding speed under different HD settings. We test the speed with a Nvidia-A100 80G. With the results in Table 10, we have three observations: 1) Prefix encoding time increases linearly with the number of prefix tokens. 2) Decoding speed remains relatively constant, regardless of prefix length, thanks to optimizations on transformers from research communities and companies, including kv-cache and flash-attention. 3) When generating 2048 tokens, total inference time usage is nearly identical across HD9 to HD55, as encoding time is much smaller. Based on the above analysis, we believe the inference efficiency of our model is acceptable. Besides, we believe some targeted designs can further improve efficiency, while our paper focuses on enabling LVLM to understand high-resolution images with a general and effective solution, and we would leave the efficiency exploration in future work.

# 5 Conclusion

In this paper, we propose the InternLM-Xcomposer2-4KHD that exceeds the performance of previous open-source models on OCR-related tasks and also achieves competitive results on general-purpose LVLM benchmarks. Thanks to our dynamic resolution and automatic patch configuration, our model supports a maximum training resolution of up to 4K HD. We also integrate a global view patch to support the macro understanding and a learnable newline token to handle the various input image resolutions. Our model's performance continues to improve as the training resolution increases for HD-OCR tasks. Notably, we do not observe any performance saturation even for the 4KHD setting, and we have not explored the upper bound due to the computational burden increasing with higher-resolution inputs. In future work, we plan to explore efficient solutions for accurate LVLM training and inference, enabling our model to handle even higher resolutions while maintaining computational efficiency.

# 6 Acknowledgment

This project is funded in part by Shanghai Artificial Intelligence Laboratory, the National Key R&D Program of China (2022ZD0160201), the Centre for Perceptual and Interactive Intelligence (CPII) Ltd under the Innovation and Technology Commission (ITC)'s InnoHK. Dahua Lin is a PI of CPII under the InnoHK.

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

## A  More examples chat with our InternLM-XComposer2-4KHD

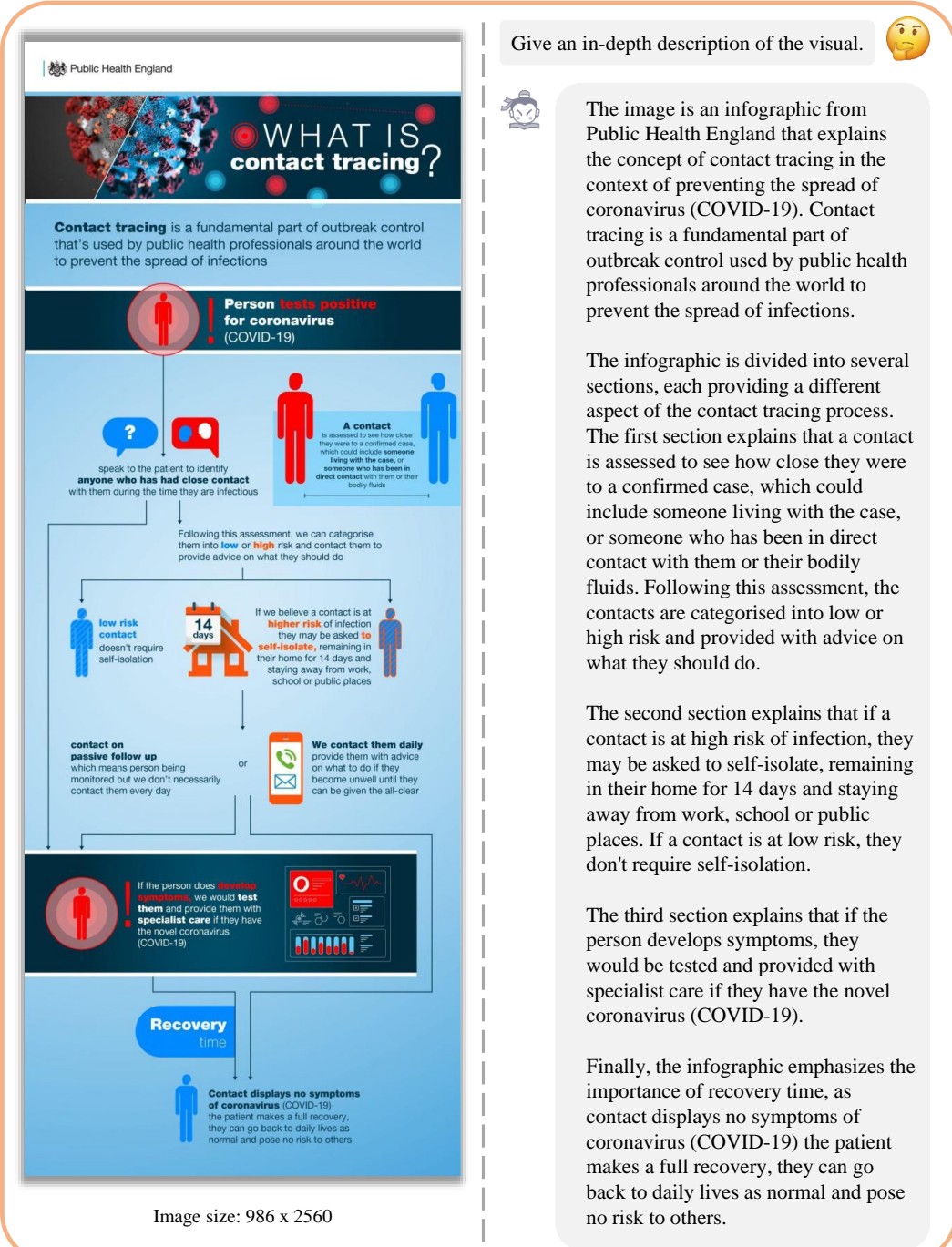

Give an in-depth description of the visual.

The image is an infographic from Public Health England that explains the concept of contact tracing in the context of preventing the spread of coronavirus (COVID-19). Contact tracing is a fundamental part of outbreak control used by public health professionals around the world to prevent the spread of infections.

The infographic is divided into several sections, each providing a different aspect of the contact tracing process. The first section explains that a contact is assessed to see how close they were to a confirmed case, which could include someone living with the case, or someone who has been in direct contact with them or their bodily fluids. Following this assessment, the contacts are categorised into low or high risk and provided with advice on what they should do.

The second section explains that if a contact is at high risk of infection, they may be asked to self-isolate, remaining in their home for 14 days and staying away from work, school or public places. If a contact is at low risk, they don't require self-isolation.

The third section explains that if the person develops symptoms, they would be tested and provided with specialist care if they have the novel coronavirus (COVID-19).

Finally, the infographic emphasizes the importance of recovery time, as contact displays no symptoms of coronavirus (COVID-19) the patient makes a full recovery, they can go back to daily lives as normal and pose no risk to others.

Image size: 986 x 2560

Figure 5: **Chat with InternLM-XComposer2-4KHD**.

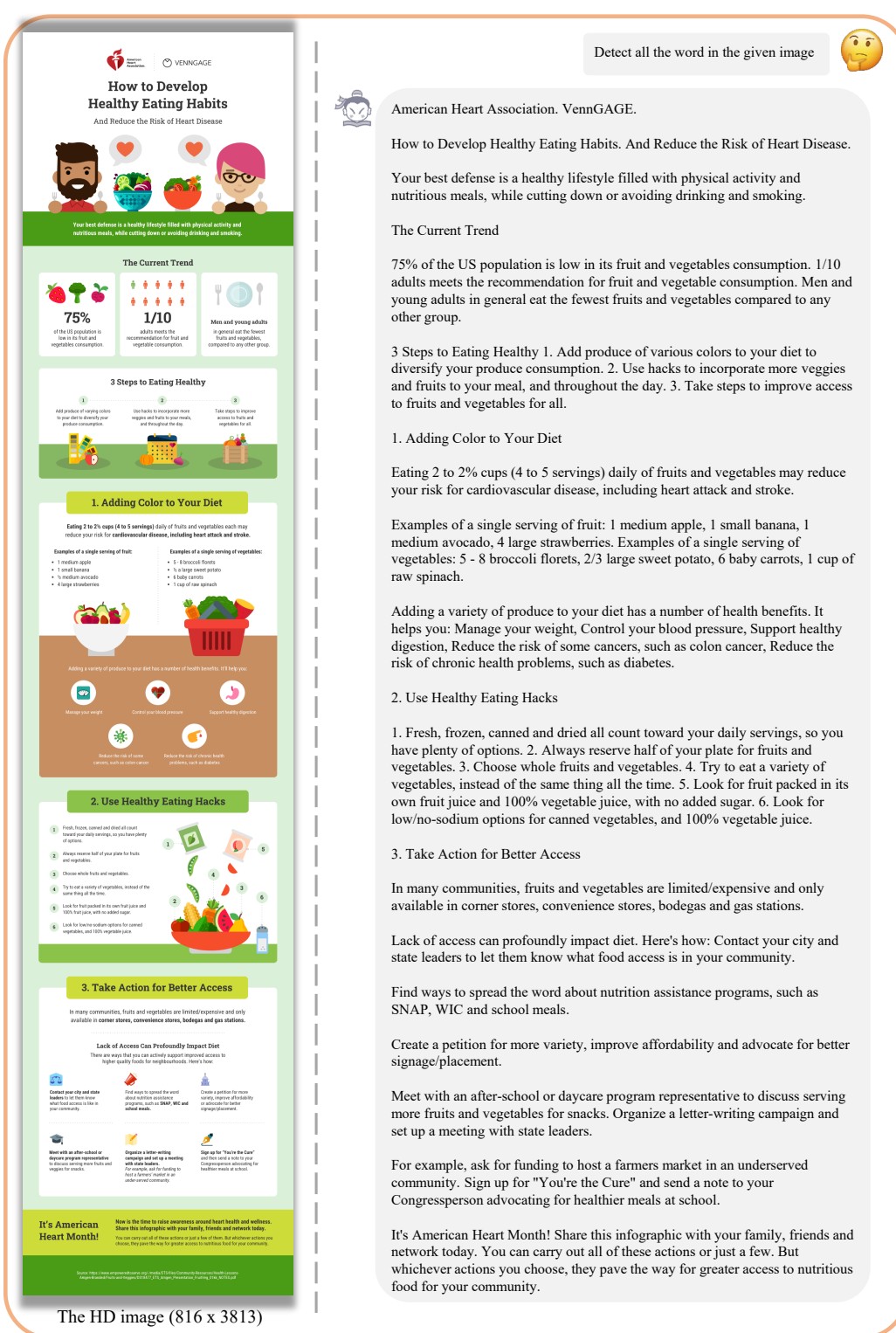

The HD image (816 x 3813)

**Detect all the word in the given image**

American Heart Association. VennGAGE.

How to Develop Healthy Eating Habits. And Reduce the Risk of Heart Disease.

Your best defense is a healthy lifestyle filled with physical activity and nutritious meals, while cutting down or avoiding drinking and smoking.

The Current Trend

75% of the US population is low in its fruit and vegetables consumption. 1/10 adults meets the recommendation for fruit and vegetable consumption. Men and young adults in general eat the fewest fruits and vegetables compared to any other group.

3 Steps to Eating Healthy 1. Add produce of various colors to your diet to diversify your produce consumption. 2. Use hacks to incorporate more veggies and fruits to your meal, and throughout the day. 3. Take steps to improve access to fruits and vegetables for all.

1. Adding Color to Your Diet

Eating 2 to 2% cups (4 to 5 servings) daily of fruits and vegetables may reduce your risk for cardiovascular disease, including heart attack and stroke.

Examples of a single serving of fruit: 1 medium apple, 1 small banana, 1 medium avocado, 4 large strawberries. Examples of a single serving of vegetables: 5 - 8 broccoli florets, 2/3 large sweet potato, 6 baby carrots, 1 cup of raw spinach.

Adding a variety of produce to your diet has a number of health benefits. It helps you: Manage your weight, Control your blood pressure, Support healthy digestion, Reduce the risk of some cancers, such as colon cancer, Reduce the risk of chronic health problems, such as diabetes.

2. Use Healthy Eating Hacks

1. Fresh, frozen, canned and dried all count toward your daily servings, so you have plenty of options. 2. Always reserve half of your plate for fruits and vegetables. 3. Choose whole fruits and vegetables. 4. Try to eat a variety of vegetables, instead of the same thing all the time. 5. Look for fruit packed in its own fruit juice and 100% vegetable juice, with no added sugar. 6. Look for low/no-sodium options for canned vegetables, and 100% vegetable juice.

3. Take Action for Better Access

In many communities, fruits and vegetables are limited/expensive and only available in corner stores, convenience stores, bodegas and gas stations.

Lack of access can profoundly impact diet. Here's how: Contact your city and state leaders to let them know what food access is in your community.

Find ways to spread the word about nutrition assistance programs, such as SNAP, WIC and school meals.

Create a petition for more variety, improve affordability and advocate for better signage/placement.

Meet with an after-school or daycare program representative to discuss serving more fruits and vegetables for snacks. Organize a letter-writing campaign and set up a meeting with state leaders.

For example, ask for funding to host a farmers market in an underserved community. Sign up for "You're the Cure" and send a note to your Congressperson advocating for healthier meals at school.

It's American Heart Month! Share this infographic with your family, friends and network today. You can carry out all of these actions or just a few. But whichever actions you choose, they pave the way for greater access to nutritious food for your community.

Figure 6: **Chat with InternLM-XComposer2-4KHD**.

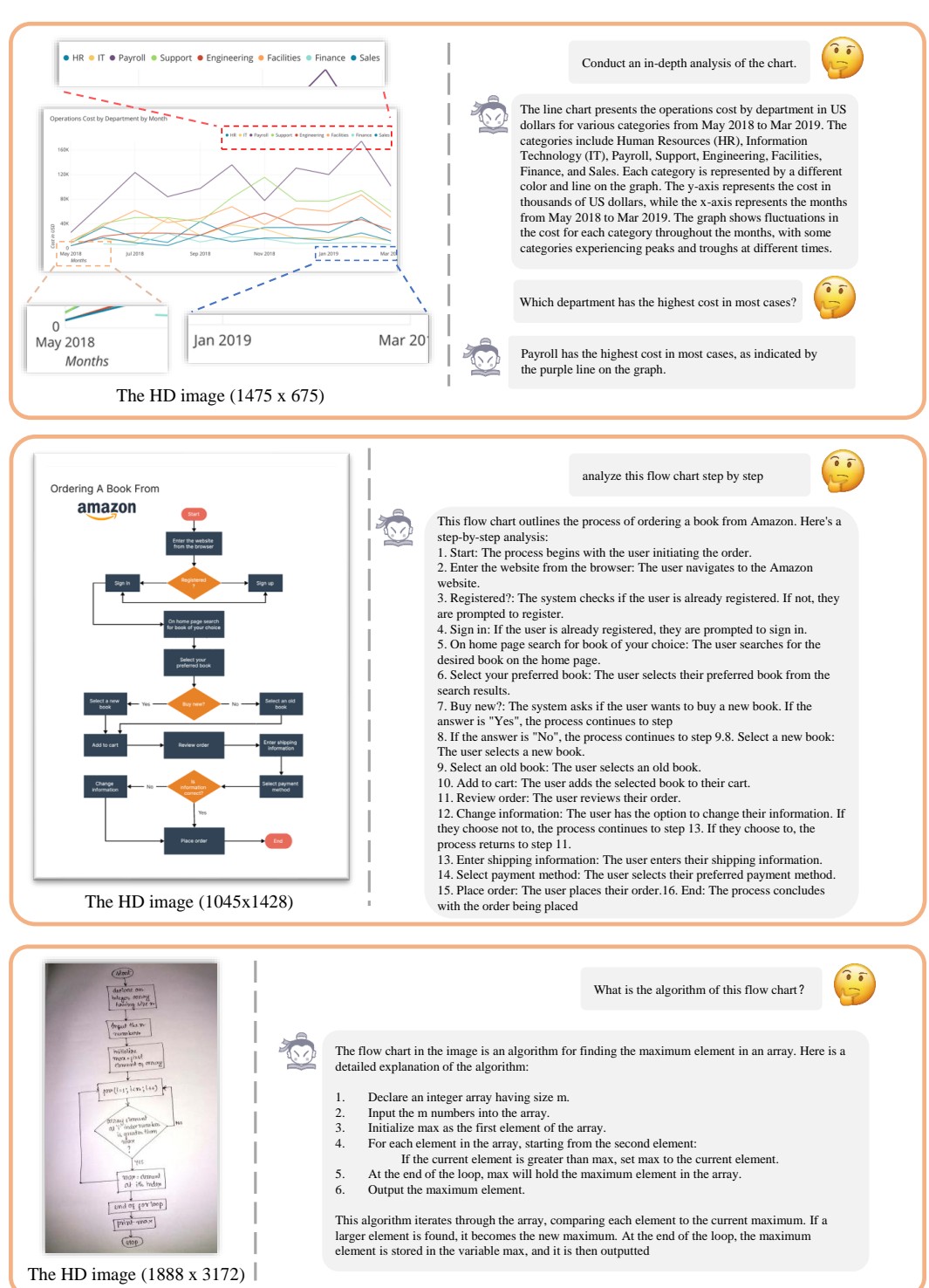

Figure 7: **Chat with InternLM-XComposer2-4KHD**.

## B  Experiment Details

### B.1  Data sampling strategy.

We maintain different dataloaders for image-text data and pure text data, sampling them in a weighted manner. The pure language dataloader is sampled with a fixed weight of $0.1$ and the image-text dataloader is sampled with a weight of $0.9$.

Within each dataloader, the training data comes from multiple sources, and we sample them in a weighted manner. In detail, for $K$ datasets and the data number of the $k_{th}$ dataset is $n_k$, its weight $w_k$ is $min(100, n_k//1000)$ and the normalized weight is $\hat{w}_k = w_k/\sum w_k$. For each $get\_data$ operation within the dataloader, we use weighted sampling to choose a dataset and randomly choose a training sample from it.

## C  Broader Impacts

On the positive side, our model introduces a novel solution that significantly enhances the ability of large language models to comprehend high-resolution images. This innovative approach is expected to be highly beneficial for the research community. It paves the way for new explorations and discoveries in the field of MLLM and image processing. The potential applications of this model are vast.

Moreover, we have plans to make our model open-source. By sharing our work with the public, we aim to foster an environment of shared learning and progress. Users, researchers, and developers can utilize our model, adapt it to their needs, and even contribute to its further development. This open-source approach will accelerate the pace of innovation and bring about more rapid advancements in the field.

On the negative side, like any powerful tool, there is a potential for misuse of our model. There is a risk that individuals with malicious intent may exploit the capabilities of the model for unethical or harmful purposes. This is a challenge that we, as researchers, must acknowledge and address. However, it is also crucial for users and the wider community to use our model responsibly and ethically. We believe that through collective effort, we can mitigate these risks and ensure that the benefits of our model are realized while minimizing potential harm.

## D  Limitation

In this paper, we study the influence of training resolution in a wide range, from HD-9 to 4KHD (HD-25 + HD-55). Our results show that high-resolution OCR-related tasks rely on the training resolution heavily, and get significant performance gains with increased resolution. Till our largest setting 4KHD, its gain is not saturated. If we keep increasing the resolution, it may get better results. However, due to computational constraints, we failed to fully explore the potential improvements from further increasing training resolution, and we have to leave it as future work.

