# OpenReview forum: "InternLM-XComposer2-4KHD: A Pioneering Large Vision-Language Model Handling Resolutions from 336 Pixels to 4K HD"
_NeurIPS.cc/2024/Conference — NeurIPS 2024 poster_

### Official Review · Reviewer_339k · 2024-06-12

**Soundness:** 4
**Presentation:** 3
**Contribution:** 3
**Rating:** 5
**Confidence:** 3

**Summary:**

This paper introduces a large vision-language model named InternLM-XComposer2-4KHD, which is designed to process images up to 4K HD resolution. The model incorporates a dynamic resolution adaptation mechanism that maintains the aspect ratios of input images, thus enhancing the model's ability to understand complex image-text interactions. This approach allows for scalable resolution adjustments during training, supporting a wide range of resolutions. The model even outperforms well-known GPT-4V and Gemini Pro in 10 out of 16 benchmarks.

**Strengths:**

1. This paper introduces a substantial enhancement in the field of large vision-language models. It demonstrates significant performance improvements across multiple benchmarks, outperforming well-known models such as GPT-4V and Gemini Pro, thereby potentially exerting a broad impact on the field.

2. The dynamic resolution design adopted in this study is particularly noteworthy. It enables the model to adapt to a variety of image resolutions while preserving critical detail.

3. The writing in the paper is commendable, with informative charts and examples that effectively showcase the model's capabilities.

**Weaknesses:**

1. The paper frequently mentions training with 4K resolution, yet it lacks optimizations for accelerating high-resolution training. The heavy reliance on substantial computational resources limits the broader applicability of the proposed methods.
2. There is a lack of analysis regarding the inference efficiency of the model. While the paper validates that high-resolution training significantly enhances performance for text-based vision-language tasks, it does not adequately address the computational costs incurred by high-resolution inputs. This omission makes it challenging to assess whether the additional computational expenses are justified by the performance gains.
3. The introduction repetitively emphasizes adaptability to 4K resolution without providing clear problem statements or motivational examples. For instance, the discussion on "variability in patch layouts" (Line 52) lacks an intuitive explanation, and it would benefit from specific examples illustrating this variability. Besides, the paper highlights at Line 56 a critical aspect concerning the model's performance at resolutions beyond 4K with 'even when...'. It remains unclear whether this suggests that traditional methods trained at 4K resolution might fail to gain additional performance at lower resolutions. Clarifying this ambiguity is crucial for understanding the challenges the study addresses, reducing the need for readers to infer details solely from experimental results.
4. This work notably advances vision-language understanding by enhancing traditional OCR capabilities. However, it prompts inquiries about the model's proficiency with complex visual content. The examples presented do not conclusively demonstrate the model's ability to accurately interpret intricate visuals such as hand-written mathematical formulas, circuit diagrams, or chemical structures. These types of content demand detailed parsing for complex problem-solving, an aspect not fully illustrated by the simpler cases in the appendices. While these considerations may exceed the current study's scope, they do not impact the overall evaluation for acceptance.

**Questions:**

1. The substantial computational resources required for training at 4K resolutions and the lack of detailed analysis on inference efficiency are main concerns.
2. The paper needs clearer problem statements and motivational examples.
3. The paper demonstrates enhancements in traditional OCR capabilities but leaves some uncertainty regarding the model's ability to process more complex visual content. If the authors can offer further insights or evidence, it would significantly strengthen the paper’s impact and might merit a higher evaluation.

**Limitations:**

The paper addresses its limitations and potential negative societal impacts in the appendices. The explanation of limitations lacks depth. For more detailed concerns, please refer to the questions section.

---

> ### Author Rebuttal · Authors · 2024-08-07
>
> ### Q1: The paper frequently...
> A1: Thanks for your valuable comments.
> 1. Enabling LVLM to understand high-resolution images remains a challenging and open problem in the field. The primary goal of our paper is to explore a general and effective strategy for high-resolution image understanding LVLMs, and further study how the image resolution influences LVLM performance.
> 2. Efficiency is an important under-explored topic, but in most cases, there is a trade-off between performance and efficiency. In this work, we focus on pushing the performance upper bound without introducing additional variables, and we would leave the study on efficiency in future work.
> 3. Our proposed weighted multi-dataloader sampling strategy (Please see Appendix B.1) is designed to optimize the training process. By enabling each GPU to process batches with either high-resolution images only or low-resolution images only, we reduce the need for padding tokens to handle unbalanced sample lengths caused by varying image sizes. This approach speeds up the training process, making it more efficient and scalable
>
> ### Q2: lack of analysis regarding the inference efficiency...
>
> A2: The model inference process consists of two stages: encoding the prefix (model input) and autoregressively decoding new tokens (model output). Correspondingly, the inference efficiency considers two parts: time to encode the prefix and speed to decode each token.
> Here we report the prefix encoding time and per-token decoding speed under different HD settings. We test the speed with a Nvidia-A100 80G.
> With the results below, we have three observations
> 1. Prefix encoding time increases linearly with the number of prefix tokens.
> 2. Decoding speed remains relatively constant, regardless of prefix length, thanks to optimizations on transformers from research communities and companies, including kv-cache and flash-attention.
> 3. When generating 2048 tokens, total inference time usage is nearly identical across HD9 to HD55, as encoding time is much smaller.
>
> Based on the above analysis, we believe the inference efficiency of our model is acceptable
>
> We believe some targeted designs can further improve efficiency, but as we discussed in A1, our paper focuses on enabling LVLM to understand high-resolution images with a general and effective solution, and we would leave the efficiency exploration in future work.
>  HD | image tokens | prefix encoding time | per-token decoding speed | time to generation 2048 new tokens
> -|-|-|-|-
>  HD9  | 1440| 0.2845| 0.0982| 201.4
>  HD16 | 2448| 0.3966| 0.0983| 201.7
>  HD25 | 3744| 0.5513| 0.0981| 201.5
>  HD55 | 8064| 1.1272| 0.0983| 202.4
>
> ### Q3: The introduction repetitively emphasizes......
> A3: Sorry for the confusion.
> 1. Regarding the variability in patch layouts, we illustrate this challenge with Fig.2 in the rebuttal PDF. Images a and b are distinct, yet when flattened into a one-dimensional input sequence, they become identical inputs (Fig. 2(c)) for the LLM. In practice, images also exhibit diverse aspect ratios, which can lead to confusion in object positioning. To address this, it is crucial for LVLM to accurately comprehend the 2D structure. We propose two core designs to achieve this: Global-Local Format and Patch Layout Indicator. The global image provides a concise overview, while the indicator precisely conveys the 2D structure to the model.
>
> 2. For 'Even when' in Line 56, this is an inference drawn from observations in Fig. 4 (model performance is not saturated even with HD55) and Table 6 (model performance improves with higher inference resolution) in our paper. We provide quantitative results on a challenging subset of InfoVQA to support this. Specifically, we selected images with resolutions exceeding 1680x3696 (HD55) pixels from the InfoVQA validation set, resulting in a 15.6% subset. We evaluated IXC2-4KHD's performance under HD55 and HD65 settings. The results, presented below, demonstrate that even though the model is trained with the 4KHD setting, it still benefits from larger inference resolutions when applied to images larger than the training cases.
>  InfoVQA HD55 subset | 55    | 65
> -|-|-
>  HD55| 70.41 | 71.64
>
> 3. For the 'methods trained at 4K resolution might fail to gain additional performance at lower resolutions,' problem, we observe that when comparing MME (at most 512x512) and MMBench (at most 512x512) results between HD9 and HD25 in Fig. 4, the performance is nearly identical. This suggests that model performance saturates when the training resolution exceeds the image's original resolution, and increasing the training resolution yields no further gain.
> In contrast, when the training resolution is smaller than the image's original resolution, model performance continues to improve with increasing training resolution, as seen in DocVQA (at most 5367×7184) and InfoVQA (at most 2921×24978) in Fig. 4.
> These observations underscore the necessity of our goal: exploring a general and effective strategy to enable LVLM to understand images across a broad range of resolutions, from 336 pixels to 4KHD.
> ### Q4: It prompts inquiries....
>
> A4: We appreciate your insightful suggestion and have investigated this issue using the college-level benchmark MMMU. Specifically, we curated a subset of high-resolution images (with resolutions exceeding 1000x1000 pixels) from the MMMU validation set, which accounts for approximately 10.5% of the dataset. We then compared the performance of IXC2 and IXC2-4KHD on this subset.
> The results demonstrate a notable improvement in performance by IXC2-4KHD, indicating its capability to not only handle dense OCR tasks on large images but also tackle complex, college-level tasks that require high-resolution input.
> Additionally, we have included two exemplary cases in the rebuttal PDF Fig.3, showcasing our model's ability to solve intricate reasoning problems based on the visual information provided.
> Model |  MMMU-val HD subset
> -|-
>  IXC2      | 46.32
>  IXC2-4KHD | 52.63

---

> > ### Author Response · Authors · 2024-08-12
> >
> > Dear Reviewer 339k,
> >
> > Thank you for the time and effort you have dedicated to reviewing our submission. Your appreciation for our work, along with the questions and suggestions you provided, has greatly helped us improve the quality of our work. We hope we have addressed the concerns raised in your initial reviews.
> >
> > As the author-reviewer discussion period for NeurIPS 2024 only has two days, please let us know if you require additional information or clarification. We are ready to engage in further discussions to enhance and elevate our work.
> >
> > Best regards and thanks,
> >
> > Paper 3325 Authors

---

> > ### Comment · Reviewer_339k · 2024-08-13
> >
> > Thanks for the response, which has addressed most of my concerns. Since I am not an expert in this field, I carefully read other reviewers' comments and the literature mentioned by Reviewer kpFC. I noticed that the methods in this paper have many similarities to existing methods. Therefore, the paper should clarify these similarities in the method section. Considering that these designs are not original in this paper, I have to adjust my score accordingly.

---

### Official Review · Reviewer_kpFC · 2024-07-11

**Soundness:** 3
**Presentation:** 3
**Contribution:** 2
**Rating:** 3
**Confidence:** 5

**Summary:**

The paper presents InternLM-XComposer2-4KHD, a high-resolution MLLM that performs better than existing methods on various benchmarks.

**Strengths:**

1. The paper is well written and easy to understand.
2. The performance of  InternLM-XComposer2-4KHD is good, which is valided on various benchmarks.

**Weaknesses:**

1. The paper has limited novelty. The high-resolution solution is not new. Starting from Monkey, there are a lot of works that adopt the similar idea, e.g., LLaVA-NeXT and InternVL-1.5.  Compared to these methods, the technical contribution of the paper is trivial.


2. From the experimental results, I cannot tell whether the proposed high-resolution solution is better than other solutions. Since the setting of the paper are not comparable to others,  it seems that the data plays a more significant role. Maybe more comparisons or ablations should be provided.


3. There still be a lot of MLLMs that have similar settings to  IXC2-4KHD, e.g., InternVL-1.5.  Why not compare with these models in Tab 3-4.

4. The implementation details are not clear. It's still confusing for me that how much data is used for training.   In Tab 3, the previsous SoTA LLaVA-NeXT uses less than 2M data. So It's not clear whether the comparison is fair.

**Questions:**

My main concern is the novelty and the experimental setting. The main idea has been widely explored in previous works.

**Limitations:**

See weakness.

---

> ### Author Rebuttal · Authors · 2024-08-07
>
> ### Q1: Limited novelty...
>
> A1: Thanks, we would like to highlight the novelty of our work in the following aspects:
> 1. As discussed in the related work (Lines 95-101), our approach is the first to address the challenges and propose solutions for handling variability in image feature patch layouts, enabling effective training with dynamic high-resolution images.
> 2. Our method introduces a dynamic resolution and automatic path configuration strategy, allowing for seamless scaling from 336 pixels to 4K resolution - a range that exceeds previous approaches. In contrast, existing methods for high-resolution understanding are limited to either predefined high-resolution settings or a narrow range of resolutions. For instance, Monkey and LLaVA-Next employ sliding windows and image splitting, respectively, but are restricted to segmenting images into a maximum of 6 and 4 patches, respectively. This limitation hinders their ability to handle images with extreme aspect ratios, commonly found in infographics. Our IXC2-4KHD, on the other hand, supports up to 55 local image patches, making it capable of tackling these challenging cases.
> 3. Our work also presents a thorough analysis of high-resolution strategy design, examining the impact of global image, 'newline' token, and training/inference resolution - aspects that were underexplored in previous studies. Through extensive experimentation, we have gleaned valuable insights, including:
>     - The LVLM struggles to comprehend the 2D structure of a simply flattened image, but incorporating a global image and 'newline' tokens could solve this problem with clear performance improvement.
>     - For tasks requiring high-resolution images, our model demonstrates improved performance with larger images, showing no signs of saturation even at 4KHD resolution. This suggests potential for further scaling and improved results.
>     - Conversely, for tasks insensitive to resolution, model performance plateaus as image size increases.
>
> 4. Finally, we note that InternVL-1.5 is a concurrent, unpublished work that has not undergone peer review or been accepted by any reputable conference or journal. Similarly, LLaVA-Next is a GitHub blog post, rather than a rigorously evaluated academic publication.
>
> ### Q2: Experimental... More comparisons or ablations.
>
> A2: Thanks for your valuable advice. For a fair comparison, we trained a new model with the identical architecture, training strategy, and dataset as our original model, but with one key modification: we adopted the high-resolution strategy from LLaVA-Next. We name it as IXC-LLaVA and compare it with 1) LLaVA-Next 0530 official results and 2) IXC2-4KHD under HD-9/16 setting. The results below demonstrate that IXC-LLaVA achieves promising performance across all six benchmarks, leveraging the benefits of additional training data and advanced IXC architecture design. However, it is outperformed by IXC2-4KHD HD-9, which utilizes fewer image tokens yet yields better results. This fair comparison underscores the efficiency and effectiveness of our proposed high-resolution strategy, highlighting its advantages over the LLaVA-Next approach.
>  Model Nmae         | HD Strategy | Image Tokens | Max Resolution | DocVQA  | TextVQA | ChartQA | MMBench
> -|-|-|-|-|-|-|-
>  LLaVA-Next(LLaMA3) | LLaVA-Next  | 2880         | 672x672        | 78.2    | 52.0    | 69.5    | 72.1
>  IXC-LLaVA          | LLaVA-Next  | 2880         | 672x672        | 78.9    | 71.5    | 74.1    | 77.6
>  IXC-4KHD           | HD9         | 1440         | 1008x1008      | 79.4    | 73.8    | 78.2    | 79.5
>  IXC-4KHD           | HD16        | 2448         | 1344x1344      | 84.9    | 75.7    | 80.1    | 80.2
>
> ### Q3: Compare with other MLLMs, e.g., InternVL-1.5.
>
> A3:
> Thank you for your comments.
> 1. As discussed in A1, InternVL-1.5 is not a peer-reviewed publication that has not been accepted by any conference or journal yet. Moreover, it is a concurrent work that is posted on Arxiv within one month to the NeurIPS submission deadline. Therefore, we do not feel obligated to compare our work with it at this time.
> 2. In Table 3, we have already reported the performance of InternVL-1.2, which was a previously released version.
> 3. In Table 4, as mentioned in line 214, we conducted a comprehensive comparison with open-source LVLMs of similar scale. However, InternVL-1.5 has a significantly larger model size (26B parameters) compared to our model (8B parameters), making a direct comparison unfair and meaningless.
>
> ### Q4: The implementation details are not clear...
>
> A4: Sorry for the confusion. We list the detailed data usage in the following. We use almost 3M data for the pre-training and 1.2M data for the SFT stage.
>
> | General Semantic Alignment    | ShareGPT4V-PT (1.2M), COCO(50K), Nocaps(20K), TextCaps (50K), SBU(200K)，LAION400M (1.2M), CC 3M  (0.3M) |
> |-|-|
> | World Knowledge Alignment     | Concept Data (500K)                                                                                     |
> | Vision Capability Enhancement | WanJuan (400K), Flicker(30K), MMC-Inst(100K), RCTW-17(8K), CTW(2K), LSVT(40K), ReCTs (20K), ArT (6K)    |
>
> | Caption| ShareGPT4V(100K), COCO (50K), Nocaps(5K)|   |   |   |   |   |   |
> |-|---|---|---|---|---|---|---|
> | General QA| VQAv2(100K), GQA (100K), OK-VQA(10K), VD(100K), RD(2K), VSR(5K)|   |   |   |   |   |   |
> | Science QA| AI2D(15K), SQA(10K), TQA(6K), IconQA(10K)|   |   |   |   |   |   |
> | Chart QA| DVQA(30K), ChartQA(7K), ChartQA-AUG(21K)|   |   |   |   |   |   |
> | Math QA| MathQA (25K), Geometry3K (2K), TabMWP (20K), CLEVR-MATH (10K)/Super (10K)|   |   |   |   |   |   |
> | World Knowledge QA | A-OKVQA (17K),KVQA (5K), ViQuAE (3K)|   |   |   |   |   |   |
> | OCR QA| TextVQA (20K), OCR-VQA (10K), ST-VQA (26K)|   |   |   |   |   |   |
> | HD-OCR QA| InfoVQA (24K), DocVQA (40K)|   |   |   |   |   |   |
> | Conversation| LLaVA-150k (100K), LVIS-Instruct4V (100K), ShareGPT-en&zh + InternLM-Chat (100K) |   |   |   |   |   |   |

---

> ### Comment · Reviewer_kpFC · 2024-08-09
>
> I appreciate authors' response, which partly addresses my concerns.  However, I still think that the novelty and contribution of this paper are still below the acceptance bar of NeurIPS.
>
> Expect for the  patch layout,  all designs of the paper exist in recent methods. For example, The global-local format and the patch division is proposed in MonKey. The Newline Indicator has been adopted  in many works like LLaVA-NeXT and LLaVA-UHD.  By the way, one thing that might be unacceptable is that the authors did not even mention these methods in their methods, which might mislead other reviewers.
>
> Most importantly, the authors attribute the high-resolution advantages to their patch layout, which is not rigorous. IXC2-4KHD is able to handle such high-resolution input not because of its patch layout, but because the token merge strategy can reduce the number of tokens so that it can be trained with a maximum token window of 4096.  This explain why Monkey and LLaVA-Next are restricted to 4-6 patches. Unfortunately, the token merge strategy is not firstly proposed in the paper, so I still doubt the novelty of the paper.
>
> Based on this concern, I think InternVL-1.5, which also applies the same token merge strategy,  is a good comparable method to prove the patch layout design.   I also understand that InternVL-1.5 may be  a concurrent work to this paper. So, maybe the authors should strictly control the same token merge strategy to prove its patch layout design.
>
> Overall, I am still inclined to reject this paper.

---

> > ### Author Response · Authors · 2024-08-09
> >
> > Thanks for your comments. We are delighted that our rebuttal has solved part of your concerns. For the newly posted comments, we shall clarify them as follows:
> >
> > ### **1. The key contribution and novelty of IXC2-4KHD lie in successfully enabling LVLM to understand images across a vast range of resolutions (336~4K).**
> > The mentioned related works are impressive that have been discussed in the related work section of our submission. It is important to note that they are explored on limited resolutions (i.e., Monkey at 1344x896, LLaVA-Next at 672x672, and LLaVA-UHD at 672x1088). Therefore, they are not able to answer the question of how to let LVLM tackle an extreme diversity of image size and aspect ratios. Moreover, IXC2-4KHD examines the impact of global image, 'newline' token,and other factors in diverse resolutions, revealing insights into their effectiveness in a unified framework for understanding dynamic high resolutions— insights not explored previously. In the final version, we will talk more about the detailed designs in Monkey, LLaVA-Next, and LLaVA-UHD and discuss our insights.
> >
> > ### **2. We would like to clarify two misunderstandings:**
> > (1) First, regarding the training of IXC2-4KHD, it was not limited to a maximum token window of 4096. Instead, IXC2-4KHD was trained with up to 8737 image tokens, which is about twice the size of 4096 tokens, as illustrated in Figure 4 of the main paper.
> >
> > (2) Second, it is our proposed dynamic patch layout strategy, rather than the token merge strategy, that contributes to the high-resolution advantages. It is worth noting that Monkey also employed a token resampler, resulting in 256 tokens for each patch (up to 16 patches with a 4096 window). Therefore, it is not the maximum token window that limits Monkey from scaling up its resolution. In summary, it is not the token merge strategy that enables high-resolution understanding.
> >
> > ### **3. We are glad the reviewer agrees that InternVL-1.5 is a concurrent work with IXC2-4KHD and should not be used to dispute our work.**
> >  We notice that InternVL-1.5 employs a similar token merge strategy, where we concatenate the nearby 4 tokens in channels, and InternVL-1.5 adopts pixel shuffle. Generally, these two designs should not make noticeable performance differences.

---

> > > ### Comment · Reviewer_kpFC · 2024-08-09
> > >
> > > Thanks for authors' further response. I don't think that there are any misunderstanding. 4096 is the maximum context window of vicuna used in LLaVA-NexT, which is the main bottleneck for its training with more visual tokens, e.g., 25 patches or more. This paper adopts InternLM2 with a longer context window, which of course supports larger resolution.
> > >
> > > Back to my concernes,  the patch division is no obviously different to monkey and llava-next. I cannot agree that increasing the  maximum number of patch from 4 patches to 55 patches  will be a big contribution to the community.  Expect for this, other contributions highlighted in the paper are all present  in recent methods.

---

> > > > ### Author Response · Authors · 2024-08-10
> > > >
> > > > Thanks for your valuable feedback. We hope the following explanations may further answer your concerns:
> > > >
> > > > 1. Indeed, the context window of Vicuna is 4096 tokens. In the meantime, the LLaVA-NexT team has officially released their LLaVA-NexT-Mistral-7B model, which utilizes Mistral-based LVLMs and features a 32K context window. However, the challenge of high-resolution understanding remains unsolved in LLaVA-NexT-Mistral-7B, indicating that handling diverse resolutions up to 4K is still non-trivial.
> > > > &nbsp;
> > > > 2. We would like to highlight that IXC2-4KHD contributes new insights to the community, enabling LVLMs to handle diverse resolutions from 336 to 4K. These insights are crucial for bridging the significant performance gap between the research community and proprietary, closed-source models. For example, IXC2-4KHD delivers a 10.8% gain on ChartQA and a 17.9% improvement on InfoVQA. For the first time, we demonstrate that LVLMs with only 8B parameters can also compete with GPT-4V on various high-resolution understanding benchmarks.
> > > > &nbsp;
> > > > On the technical aspect, we have acknowledged the previous works on patch division (Lines 43-44 in the paper) and would like to highlight our innovations in dynamic resolution with automatic patch configuration design, including our training and inference strategies. These innovations mitigate the critical obstacles posed by the scarcity of high-resolution training data and provide key insights into training and inference across extremely diverse resolutions, which have not been fully addressed in previous works yet.
> > > >
> > > > In summary, we sincerely appreciate your comments. However, we would like to clarify that the contribution of IXC2-4KHD encompasses much more than merely increasing the number of patches from 4 to 55. We are genuinely grateful for any further discussion or additional comments you may have.

---

### Official Review · Reviewer_Lqsg · 2024-07-14

**Soundness:** 4
**Presentation:** 3
**Contribution:** 4
**Rating:** 8
**Confidence:** 5

**Summary:**

This paper aims to explore the high resolution scenes of multimodal large language models. The authors find that the performances are largely improved when the model is equipped with 4K resolution. This finding will greatly inspire subsequent research works, which is of great value to the research community. The experimental results in the popular multimodal datasets, such as MME and MathVista, are impressive. The proposed dynamic resolution and automatic patch configuration will become a common approach, due to its simplicity and effectiveness.

**Strengths:**

1. The motivation sounds good. High resolution is the inevitable development direction of multimodal large language models, and we urgently need to know what effects high resolution will bring to model training and performance. This paper stands on a high ground.

2. The method looks simple yet effective. Dynamic resolution and automatic patch configuration could not only maintain the aspect ratio of the image, but also indirectly enlarge the resolution of the image.

3. The experimental results on many datasets are impressive. The proposed method achieves SOTA results in 6 of the 16 benchmarks with only 8B parameters.

4. The writing of this paper is good, and the structure is easy to follow.

**Weaknesses:**

1. Considering that high resolution is a very meaningful direction, it is suggested that the authors provide more details of the training to help other researchers, such as the loss figure of the training. Because training at high resolution is expensive, releasing more intermediate results from training will help researchers conduct further studies.

2. The current method seems to largely increase the number of visual tokens while increasing the resolution. One question is whether it is reasonable to use so such many visual tokens for ONE image. I would love to hear the authors' thoughts on this point. This is an open question.

3. It is suggested that the authors could enlarge the LLM parameters synchronously in subsequent studies if conditions permit, which will make the study more complete. It seems that image resolution and LLM parameters are the two most important aspects affecting model performance, in addition to the training data of course.

4. I am curious about the generalization performance of the proposed model on multimodal video evaluation dataset, such as Video-MME [1], which applies to both image MLLMs, i.e., generalizing to multiple images, and video MLLMs. The sub-figures in the split-patch strategy has some similarities with video frames. It would be great if the authors could update the results of the Video-MME into the paper.

[1] Video-MME: The First-Ever Comprehensive Evaluation Benchmark of Multi-modal LLMs in Video Analysis. Arxiv, 2024.

**Questions:**

None

---

> ### Author Rebuttal · Authors · 2024-08-07
>
> ### Q1: More training details
>
> A1: Thanks for your valuable comments, here we show the training loss in the Pretrain/SFT stage. Please refer to Figure 1 of the rebuttal pdf.
>
> ### Q2: Is it reasonable to use so such many visual tokens for ONE image?
>
> A2:
> This is a great and interesting question, and we address the reasonableness of using more tokens for HD images by considering the following:
> 1. **Can an LLM understand an image with so many tokens?** Our paper reveals that the LVLM model can effectively leverage a large number of image tokens. It captures image details from thousands of image tokens, leading to significant gains on various high-resolution benchmarks.
> 2. **Is an image really worth thousands of tokens?** The aim of our paper is to develop a universal solution for LVLMs to analyze images with varying resolutions, ranging from 336 pixels to 4K resolution. Further token compression may result in unavoidable information loss and impact our study. As discussed in Figure 4 of the main paper, we observed that more visual tokens lead to consistent performance gains on benchmarks requiring fine-detailed understanding, such as DocVQA and InfographicVQA. This implies the necessity of scaling up the image resolution for a single image. We leave the task of increasing image resolution with efficient image token numbers as a future research direction.
>
> ### Q3: The influence of  LLM parameters
>
> A3: Thanks for your valuable idea! Due to the rebuttal period limitation, there is no sufficient time for us to train a 20B LVLM. Instead, we conduct the experiment on a smaller model InternLM2-1.8B and we found a clear performance improvement with LLM parameter increasing. We believe if we keep increasing the LLM size, the model will get better performance.
>
> | Model    | LLM Size | DocVQA  | InfoVQA | TextVQA | ChartQA | MMBench | MME     | SEED  |
> |----------|------|---------|---------|---------|---------|---------|---------|-------|
> | IXC-4KHD | 1.8B | 81.7    | 54.9    | 70.1    | 72.4    | 70.6    | 1,973.0 | 70.5  |
> | IXC-4KHD | 7B   | 90.0    | 68.6    | 77.2    | 81.0    | 80.2    | 2,204.9 | 74.7  |
>
> ### Q4: Performance on Video Benchmarks
>
> A4: Thanks for your valuable comments, we tried to concatenate the video frames into a long image and finetuned it with the SFT data used in Video-ChatGPT [1]. Here we report the results on three widely used video benchmarks. As shown below, our model performs on par with GPT-4V on the challenging MME-Video, and significantly better than GPT-4V on the MVBench and MLVU.
>
> | Model    | MME-Video w/o subtitle | MVBench | MLVU  |
> |----------|------------------------|---------|-------|
> | GPT-4V   | 56.0                   | 43.5    | 49.2  |
> | IXC-4KHD | 54.2                   | 56.7    | 65.1  |
>
> [1] Video-ChatGPT: Towards Detailed Video Understanding via Large Vision and Language Models

---

> > ### Comment · Reviewer_Lqsg · 2024-08-09
> >
> > I have carefully read the rebuttal. The rebuttal solved my questions and I think the study about adaptive resolution LVLM is novel and beneficial to the research community. The results of video benchmarks also show the potential of the proposed solution.
> >
> > I also read the review comments from other reviewers, I think from both study depth and scale, it provides novel insight and understanding about how to enable LVLM to understand images with high resolution, it is distinguished from existing works such as LLaVA-Next.
> >
> > In conclusion, I would keep my score as 8, strong accept. If the author could open-source data, code, and more experiment details about this work in the future, it would be helpful for the following study.

---

> > > ### Author Response · Authors · 2024-08-11
> > >
> > > Thank you so much for appreciating the novelty and contribution of IXC2-4KHD to the community. We are also grateful for your recognition of IXC2-4KHD is distinguished from existing works. We appreciate your suggestions and will add more experimental details in our revision. Additionally, our data and code will be made public to benefit the community.

---

### Author Rebuttal · Authors · 2024-08-07

We sincerely thank the efforts of all the reviewers and the AC. We are encouraged by the acknowledgment of our strong performance by  Reviewer Lqsg, Reviewer kpFC, and Reviewer 339k, and the noteworthy and effective model design by Reviewer Lqsg and Reviewer 339k. We answered all the questions in the rebuttal and attached the necessary images in the pdf. We hope our rebuttal could solve the Reviewers' concerns.

---

### Decision · Program_Chairs · 2024-09-25

**Decision:**

Accept (poster)

**Comment:**

This work proposes a novel large vision-language model capable of handling a dynamic range of resolutions from 336 to 4K pixels. To achieve this, it combines various existing design ideas, including the use of global and local patches, learned new line tokens and token merging along with new ideas for dynamic layout prediction. The proposed method is shown to outperform the existing open source methods in various VLM tasks requiring inference with high resolution images, while maintaining accuracy on general purpose VLM tasks.

Reviewers raised concerns about the novelty of the proposed ideas used to construct the method; the lack of comparisons to several recent SOTA approaches and the inference and training speed of the method. Several of the reviewers' concerns were addressed in the rebuttal, while requested comparisons are to recent and unpublished works. While the AC agrees that several of the ideas used to build the proposed method are known from prior works, they feel that the current work does employ them in a novel manner to achieve scalable and scalable inference at various image resolutions. In terms of quality this work presents a significant step forward for open source models. Since the authors have promised to release the data and code as well, it will a valuable resource for the research community to advance open source models further along the dimension of processing high resolution images and will form a strong foundation for the community to build upon. How to process higher resolution images in LVLM is a significant and important unsolved problem. Hence, all things considered the AC leans towards accepting this work and recommends that the authors incorporate the changes that they have promised in the rebuttal into their final manuscript.